# The evolution of negotiation strategies diversifies parental cooperation
Jia Zheng [1,2,3] ✉, Franz J. Weissing [3] ✉ & Davide Baldan [4,5]

Parental cooperation is not self-evident, as conflicts often arise over individual contributions. Evolutionary game theory suggests this conflict may be resolved through negotiation, where parents adjust their care level based on their partner's contribution. However, mathematical negotiation models typically predict low parental cooperation. As these models are not dynamically explicit and mostly neglect stochasticity, we employ individual-based simulations to investigate how parental negotiation strategies evolve and shape care patterns. Our results differ markedly from earlier analytical predictions. Parental negotiation strategies readily evolve, resulting in four alternative care patterns: uniparental care, sex-biased care and two types of egalitarian biparental care. Effective cooperation evolves regularly but, contrary to common expectations, always relies on a Tit-for-Tat strategy rather than parental compensation. Our study underscores that diverse cooperative patterns in animals can emerge from sex-specific negotiation strategies, even in the absence of initial sex roles and environmental variation.

Animals exhibit extraordinary variation in the amount each parent contributes to caring for their joint offspring[1]. Traditionally, this variation is classified based on the number of caregivers, encompassing biparental, uniparental, and no care. This classification, however, is not the whole story. In some systems, biparental care is relatively egalitarian, with parents sharing their workload equally[2,3], while it is strongly asymmetric in other systems, often in a sex-specific manner[3–5]. Such differences may arise due to sexual conflict over parental care[6,7]. While both parents benefit from a higher investment of their partner, it is mainly the partner that has to pay the costs for this investment. Therefore, there is a tension between the need for parental cooperation and the temptation to free-ride on the partner's effort[8,9]. This antagonistic coevolution between the two sexes ('sexual conflict') has stimulated decades of theoretical and empirical work, attempting to explain how parental care variation and parental sex roles emerge[10–16].

Many theoretical models make the simplifying assumption that each parent chooses a care strategy that is either fixed or employed only once during a brood care period. Such a strategy, which is assumed to be genetically determined, can be the probability of deserting the nest[10,17,18] or the parental effort (in terms of time, energy, and/or resources) devoted to the current brood[12–15]. The resulting two-player game is then analysed either mathematically (by conducting an ESS analysis[12,13] or applying a selection gradient approach[14,15]) or through individual-based evolutionary simulations[16,17]. This basic approach has been refined by considering

'dynamic' parental games where the parental strategies can change with the time of the season or the parental states[19,20]. Moreover, the parental game was extended in various ways, allowing the study of the evolutionary interplay of parental care decisions with other factors, such as sexual selection[14–16,21] or various sex ratios, such as the operational sex ratio (OSR) or the adult sex ratio (ASR)[14,15,22,23].

All these approaches neglect the possibility that during the brood care period each parent flexibly responds to the behaviour of the other parent. It is well known from evolutionary game theory that such responsiveness is a game changer[24,25]. For example, in the absence of responsive strategies, non-cooperative behaviour is the only evolutionarily stable outcome in the repeated Prisoner's Dilemma game, while cooperation readily evolves if responsive strategies like Tit-for-Tat are also considered[26,27]. In line with this, McNamara and colleagues[9,28] argued that, in organisms with extended periods of biparental care, parental games should be modelled as involving a series of interactions in which the parents negotiate their mutual parental effort via inherited rules that prescribe how to respond to the behaviour of the other parent. A negotiation rule might, for example, prescribe the reduction of one's own parental effort whenever the partner shows less effort than anticipated (as in the Tit-for-Tat strategy). Alternatively, it might prescribe compensatory behaviour that enhances an individual's own effort in cases where the partner exhibits a reduced effort. According to this view, models should focus on the evolution of such rules, instead of the evolution

[1]Ministry of Education Key Laboratory for Biodiversity Sciences and Ecological Engineering, College of Life Sciences, Beijing Normal University, Beijing, China. [2]Groningen Institute for Evolutionary Life Sciences, University of Groningen, Groningen, The Netherlands. [3]Institute of Ecology and Evolution, University of Bern, Bern, Switzerland. [4]Department of Biology, University of Padova, Padova, Italy. [5]Department of Biology, University of Nevada, Reno, NV, USA. ✉e-mail: jia.zheng@unibe.ch; f.j.weissing@rug.nl

of fixed levels of care[28]. This approach is indeed more realistic, as there is ample evidence that caregivers respond to each other's behaviour in nature[29–33].

When analysing the classical parental care game of Houston and Davies[12] from a negotiation perspective, McNamara and colleagues[28] showed that responsiveness does indeed make a difference. They concluded that, rather than fostering parental cooperation, the evolutionarily stable negotiation rule leads to a lower parental provisioning rate than the non-negotiated fixed level of the Houston-Davies model. The offspring are generally better off with two jointly caring parents than with a single caregiver, but under some circumstances, negotiation hampers parental cooperation to such an extent that the offspring would benefit from uniparental care[13]. These theoretical results inspired numerous empirical tests (reviewed in refs. 9,29,34,35), which led to a diversity of outcomes, including effective parental cooperation[34,36,37]. Several proposals have been made to reconcile such cooperation with negotiation theory[34,36,38]. For example, Johnstone and colleagues[36,38] proposed a new type of negotiation model where the negotiation strategies are solely based on the identity of the parent who last contributed to care. In the most basal variant of the model[36], the evolved negotiation strategy induces the parents to reciprocate parental care by taking turns in offspring provisioning over time. Although turn-taking-like behaviour has been observed in many empirical studies (reviewed in refs. 35,37), there is still debate on whether and how well the observations fit the model predictions[39,40].

Although incorporating mutual responsiveness in parental care models is a substantial step forward, negotiation theory is not yet fully solidified. Current models investigating the evolution of responsive strategies employ mathematical (or numerical) methods that are based on the detailed analysis of the two parental fitness functions. Although such analyses are enlightening, they often do not tell the whole story. To keep the mathematical analysis tractable, many simplifying assumptions must be made that can strongly affect the model outcome. For example, analyses typically take a specific type of fitness function as their point of departure, without deriving this function from biological considerations. As explained in Supplementary Text 1, the most popular fitness functions (because of their mathematical convenience) are often not consistent with ecological or demographic considerations. Moreover, Johnstone and Hinde[34] showed that stochasticity and uncertainty can be relevant, both quantitatively and qualitatively, for the evolutionary outcome. Lessells and McNamara[41] noted an inconsistency in the early negotiation models, as they did not specify the relationship between negotiation and provisioning. Taylor and Day[42] pointed out another shortcoming of the early analyses: they did not specify the degree of responsiveness in the negotiation rules. If responsiveness is allowed to evolve, a continuum of (neutrally stable) negotiation rules arises, rather than a single evolutionarily stable rule. Interestingly, simulations (whose type and assumptions are not specified in ref. 42) indicated that parental cooperation is more prevalent than suggested by the mathematical analysis.

Here, we propose to complement mathematical theory by studying the evolution of parental negotiation rules through an individual-based simulation approach. Individual-based evolutionary simulations consider two timescales: a 'behavioural' timescale, where individuals grow up, interact with other individuals and produce offspring according to inherited strategies (in our case, negotiation rules), and an 'evolutionary' timescale, where the frequency distribution of these strategies changes due to the combined action of natural selection, mutation, and genetic drift. Such an approach has several advantages. First, individual interactions must be specified in detail, preventing misinterpretations due to hidden assumptions (e.g., implicit assumptions about the separation between a negotiation and a provisioning phase[41]). Second, individual-based simulations incorporate various types of stochasticity (which can be essential for the evolutionary outcome) in a natural and realistic manner. Third, there is no need to specify fitness functions when running individual-based simulations. In the past, it has been shown repeatedly that the specification of such functions is error-prone[11,43]. More importantly, when the evolving strategies are multivariate

(as in the case of negotiation strategies), the evolutionary dynamics do not necessarily follow the fitness gradient; as a consequence, standard mathematical methods may arrive at misleading conclusions[16]. Fourth, behavioural polymorphisms readily emerge in individual-based models; such polymorphisms are easily overlooked by mathematical analyses, although they can be crucial for the evolutionary outcome[16,44]. Last but not least, even complex scenarios can be easily translated into an individual-based evolutionary model, allowing one to tailor the model to specific empirical systems and research questions[17,45].

To our knowledge, an individual-based simulation approach has only once been employed to study the evolution of negotiation rules – in a study by Quiñones and colleagues[45] that demonstrates that negotiation can be a more effective driver of cooperation than kin selection. Here, we follow their lead and investigate the evolution of negotiation rules in the context of parental care. This allows us to address a variety of research questions. Does the evolution of negotiation rules lead to a well-defined pattern of parental care, or are alternative evolutionary outcomes possible? Can asymmetric parental sex roles evolve even if the sexes are initially identical in the model? Can parental cooperation break down completely, resulting in the replacement of biparental care by uniparental care? What type of care is most effective from the offspring's perspective? Is, for example, negotiated biparental care less favourable for the offspring than unnegotiated care as the mathematical models suggest?

## Results
### Model overview

We consider an individual-based model for the evolution of parental care strategies. The individuals in our model can survive for several breeding seasons. Each individual can only breed once per season. To this end, the adult individuals can form pairs and produce a single clutch of offspring. The survival probability of the offspring increases with the parents' total food provisioning rate; for the individual parent, the survival probability to the next season is negatively affected by its own food provisioning rate. Hence, there is a conflict of interest between the parents: both profit from a high total provisioning rate, but each parent prefers the other to do most of the provisioning. Our model investigates whether and how the evolution of parental negotiation strategies can resolve this conflict. The negotiation process of our model is illustrated in Fig. 1. The offspring provisioning phase is subdivided into discrete 1-hour time periods, within which provisioning behaviour occurs on a minute-level timescale. At the start of each time period, each parent decides on the provisioning rate for the subsequent period. This decision is based on an inherited 'behavioural reaction norm,' that is, a rule telling the parent what provisioning rate to adopt in response to the perceived provisioning rate of the partner in the previous period (Fig. 1B). As shown in Fig. 1C, the combination of the two parental reaction norms leads to a sequence of parental provisioning rates that often (but not always) converges to an equilibrium (Fig. 1D). This iterative process determines the total provisioning effort of both parents and, consequently, the survival probability of the parents and their offspring. Each reaction norm is determined by two parameters: a parameter $\alpha$ that corresponds to the provisioning rate of the partner that elicits half of the maximal provisioning rate in the following time period and a parameter $\beta$ that determines the slope of the reaction norm (Fig. 1B). These two parameters are encoded by genes at an autosomal locus and transmitted in a Mendelian manner from parents to their offspring (subject to rare mutations). Some combinations of $\alpha$ and $\beta$ will correspond to reaction norms (or negotiation rules) that yield their bearers a higher reproductive success than others. Over evolutionary time, such rules will spread in the population, eventually leading to a characteristic food provisioning pattern.

### Evolution of fixed provisioning rates

As a standard of comparison, we first ran simulations for a baseline model without negotiation. The baseline model is identical to the negotiation model, with the exception that the parental provisioning rates (called $F_m$ and $F_f$ for the male and female parent, respectively) are heritable parameters

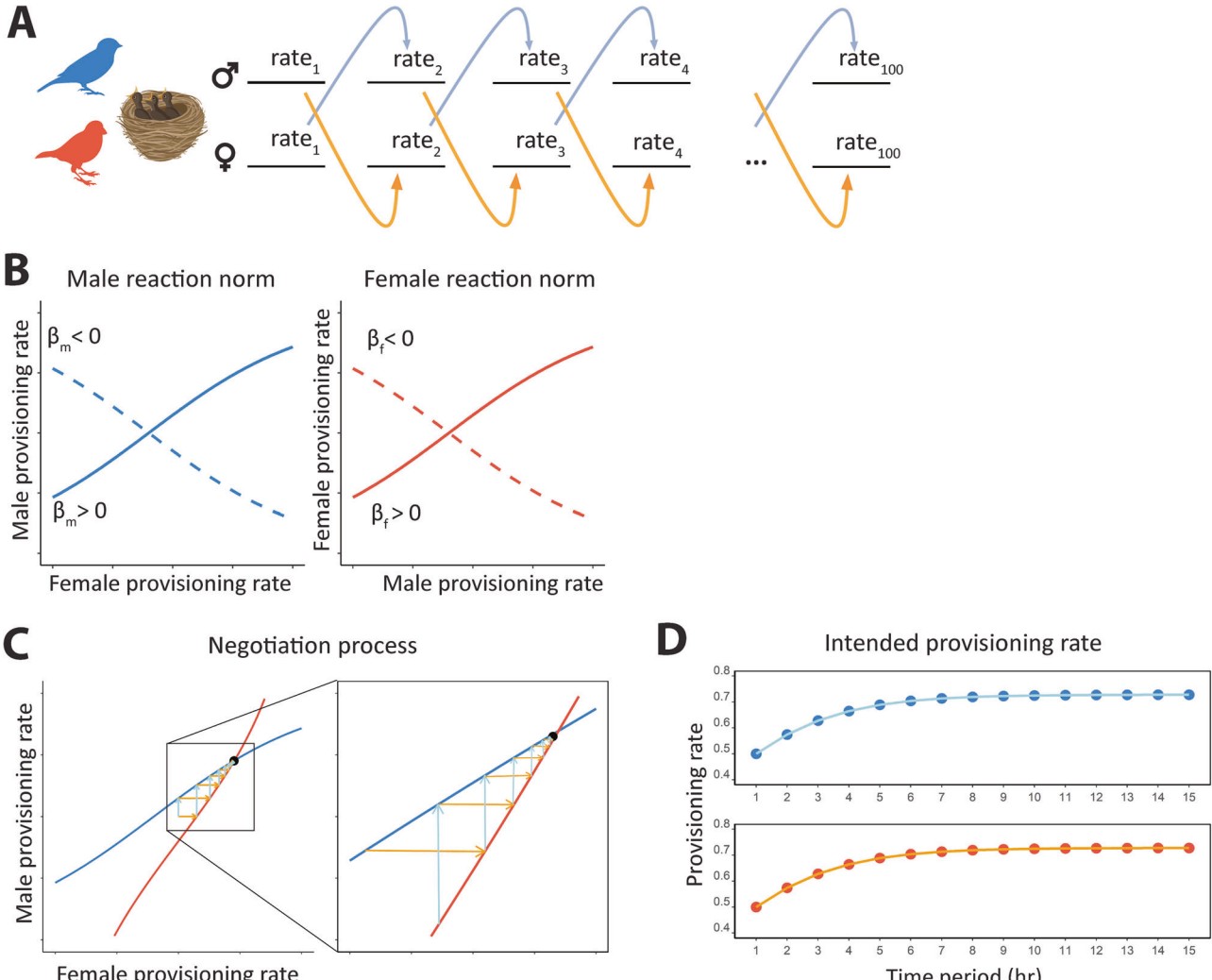

**Fig. 1 | Negotiation over parental care in repeated bouts. A** Two parents invest in a common set of offspring over a discrete number of time periods. At the start of each time period, each parent adjusts its provisioning rate in the new period to their partner's provisioning rate in the previous time period. **B** This adjustment is based on heritable behavioural reaction norms (or negotiation rules) that evolve over the generations. In our model, the reaction norms are described by logistic functions, which are determined by two heritable parameters: a parameter α determining the location of the inflexion point and a parameter β determining the slope at the inflexion point. For each sex, two example reaction norms are shown. The solid lines are increasing functions ($\beta > 0$), implying that the provisioning rate elicited by the partner increases with the partner's provisioning rate in the previous time step. The dashed lines illustrate decreasing reaction norms ($\beta < 0$) where a higher provisioning rate by the partner elicits a lower provisioning rate in the next time period. **C** The negotiation process is illustrated by overlaying the solid reaction norms of a male and a female in a single graph (thereby swapping the axes of the female reaction norm in graph **B**). For any combination of provisioning rates in the previous time period, the light blue arrow indicates the response of the male, while the yellow arrow indicates the response of the female. **D** In the example shown ($\beta > 0$ in both parents), the repeated mutual adjustment of parental provisioning rates converges monotonically to stable sex-specific provisioning rates. The figure was created in Adobe Illustrator and the vector images were taken from Adobe Stock.

that are fixed throughout an individual's lifetime (see Methods). As shown in the Supplement, all simulations rapidly converged to an evolutionary equilibrium where, for our default parameters, both parents exhibit a provisioning rate of about $F_m = F_f = 0.53$ feeds per minute, resulting in a total provisioning rate of $F_m + F_f = 1.06$. Hence, we observed the evolution of egalitarian biparental care, with no indication of alternative evolutionary outcomes.

## Alternative outcomes in the evolution of negotiation strategies

The evolutionary dynamics of the negotiation model differs markedly from that of the baseline model. Hundreds of replicate simulations (with identical simulation parameters and identical starting conditions) resulted in six alternative evolutionary outcomes, which are illustrated in Fig. 2. As shown in Fig. 3, each of the six evolutionary attractors induces a different provisioning pattern of the two parents. Figure 3A classifies the six evolutionary

attractors in terms of $(\alpha_m, \beta_m, \alpha_f, \beta_f)$, the evolved values of the parameters determining the behavioural reaction norms (interpreted as parental negotiation rules), and indicates the relative frequency of each of the six outcomes. Figure 3B illustrates how the alternative pairs of evolved negotiation strategies result in distinct provisioning patterns. We will now briefly describe these outcomes one by one.

(P1) Egalitarian biparental care. This outcome corresponds to the situation where the slopes of the evolved reaction norms of both parents are positive ($\beta_m > 0, \beta_f > 0$). It occurred in 52% of our simulations. In this case, the negotiation process (represented by the cobweb procedure in Fig. 1C) results in the rapid convergence of the parental provisioning patterns to the intersection point of the two reaction norms in the left panel of Fig. 3B(P1). In the simulation shown, this corresponds to an average provisioning rate of $F_m = 0.58$ feeds per minute for the male and $F_f = 0.64$ for the female. The actual provisioning rates of a randomly chosen breeding pair throughout the

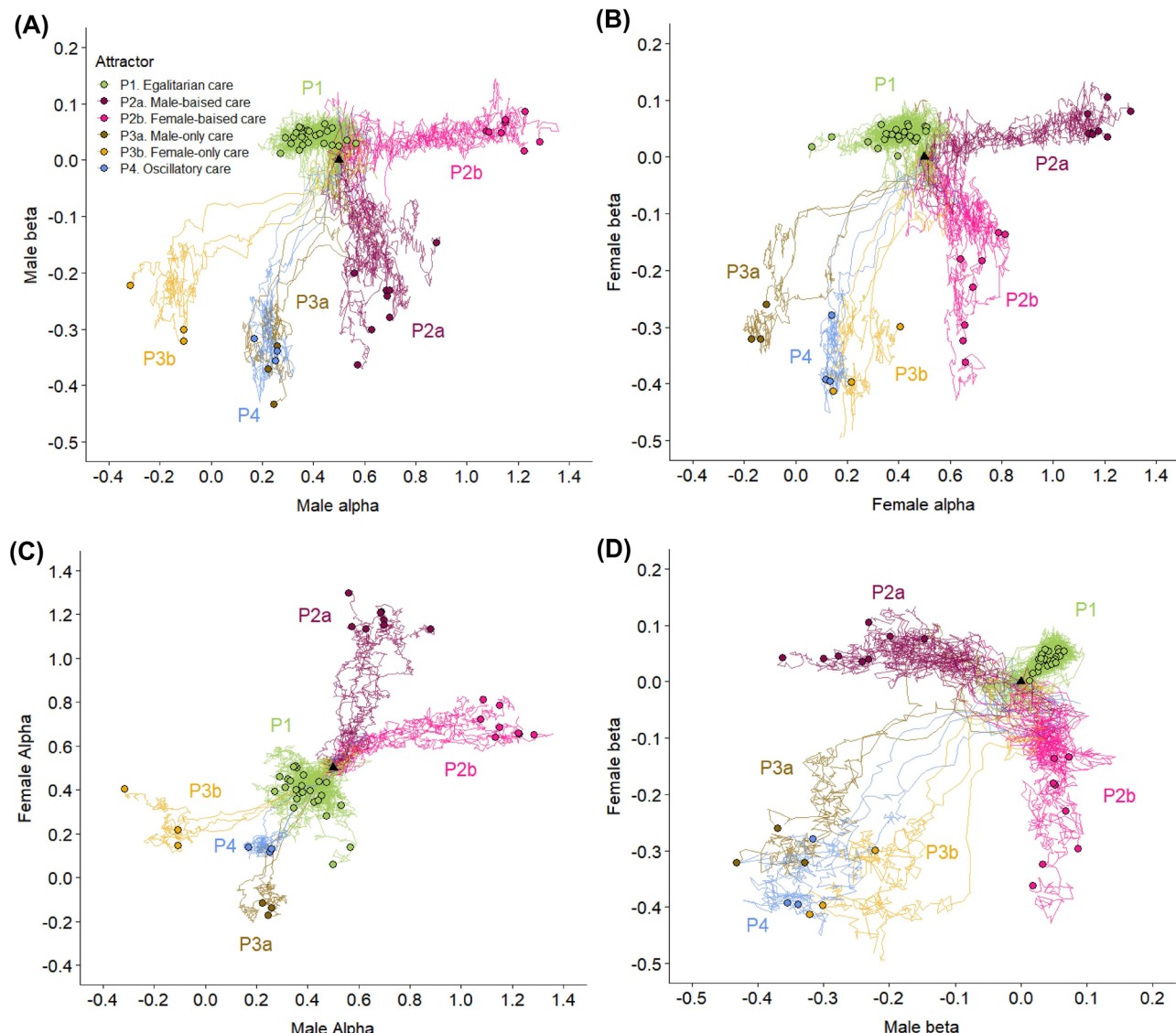

**Fig. 2 | Alternative outcomes of the coevolution of male and female negotiation strategies.** For 50 replicate simulations, the evolutionary trajectories of the heritable parameters $(\alpha_m, \beta_m)$ and $(\alpha_f, \beta_f)$ that determine the male and female negotiation strategies are shown in four panels where **A** $\beta_m$ is plotted against $\alpha_m$, **B** $\beta_f$ is plotted against $\alpha_f$, **C** $\alpha_f$ is plotted against $\alpha_m$, and **D** $\beta_f$ is plotted against $\beta_m$. All simulations were initialised at $\alpha_m = \alpha_f = 0.5$ and $\beta_m = \beta_f = 0.0$ and run for 40,000 seasons. The initial conditions correspond to a population where males and females provision their offspring at a constant rate of 0.5 and are indicated in the graphs by a black triangle. The evolved state of each simulation after 40,000 seasons is indicated by a solid dot. The 50 trajectories are representative for >1000 simulations we have run for our model: all converged to six different attractors, which in the panels are indicated by different colours. Each attractor corresponds to a different provisioning pattern (see Fig. 3): (P1) egalitarian biparental care (green), (P2a) male-biased care (purple), (P2b) female-biased care (pink), (P3a) male-only care (brown), (P3b) female-only care (yellow) and (P4) oscillatory care (blue).

breeding season fluctuate around these values because the actual number of feeds is subject to stochasticity.

(P2ab) Sex-biased care. This outcome corresponds to the situation where the slopes of the parental reaction norms evolve opposite signs, which occurred in 30% of the simulations. When the evolved reaction norm of the male is increasing while that of the female is decreasing ($\beta_m > 0, \beta_f < 0$), the provisioning pattern is female-biased (P2b), as illustrated in Fig. 3B(P2). In the simulation shown, the negotiation process results in an outcome where females provision at the maximal rate ($F_f = 1.0$ feeds per minute) while the provisioning rate of males is much lower (about $F_m = 0.35$ feeds per minute). Hence, an asymmetry in parental behaviour evolves (where one parental sex is 'exploited' by the other sex) even though our model does not include any asymmetries between the sexes. The opposite pattern, male-biased care (P2a), occurred as frequently in our simulations (15%) as female-biased care (P2b). This pattern results if the slope of the evolved negotiation

strategy of the female is positive while that of the male is negative ($\beta_m < 0, \beta_f > 0$).

(P3ab) Uniparental care. While in sex-biased care both parents participate in the provisioning of the offspring, only one parent does all the provisioning in the case of uniparental care. This outcome evolved in 12% of the simulations (6% resulting in male-only care, P3a, and 6% in female-only care, P3b). Figure 3B(P3) shows an example of male-only care, where throughout the breeding season the male provisions the offspring at the maximal rate ($F_m = 1.0$), while the provisioning rate of the female is zero ($F_f = 0.0$). Such uniparental care resulted in all simulations where the slopes of both reaction norms evolved to negative values ($\beta_m < 0, \beta_f < 0$), while the evolved values of $\alpha_m$ and $\alpha_f$ differed considerably. As indicated by Fig. 2C, male-only care results when $\alpha_m$ is substantially larger than $\alpha_f$ ($\alpha_m \gg \alpha_f$), while female-only care corresponds to the opposite relationship $\alpha_f \gg \alpha_m$.

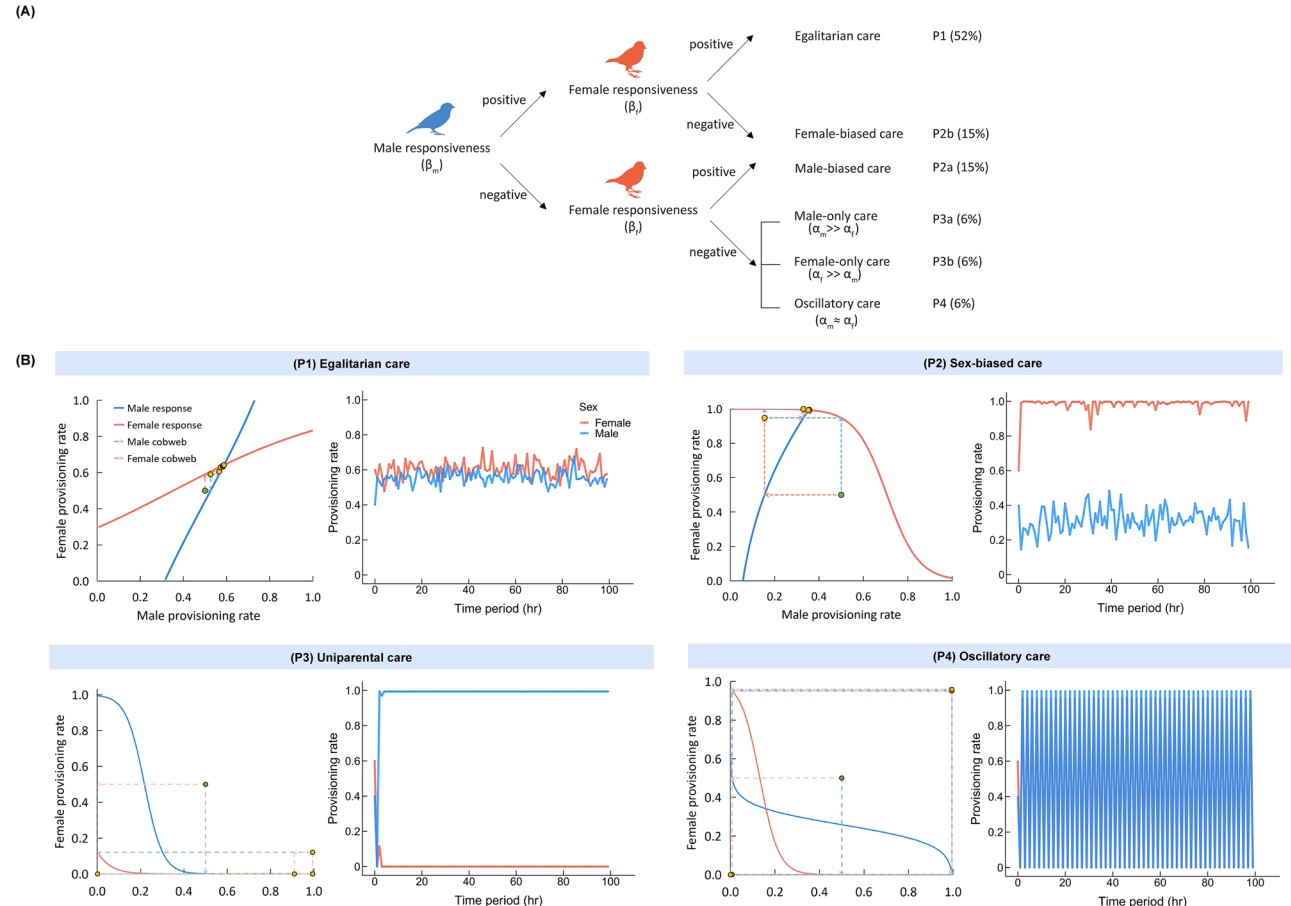

**Fig. 3 | Classification of evolved alternative provisioning patterns.**
**A** Classification of the six evolutionary attractors in terms of the evolved values $(\alpha_m, \beta_m, \alpha_f, \beta_f)$ that determine the male and female negotiation strategies. This figure was created in Adobe Illustrator and the vector images were taken from Adobe Stock. **B** The six alternative attractors induce distinct provisioning patterns that are illustrated in panels P1–P4. Each panel shows one representative simulation. The sub-panels to the left show the evolved behavioural reaction norms of the male (blue) and female (red) parent and a cobweb that indicates the resulting sequence of provisioning rates. Starting at the green dot in the centre of the cobwebs, the yellow dots represent the provisioning rates in the course of the negotiation process. The subpanels to the right show the provisioning patterns of a randomly chosen pair of parents throughout the provisioning period. In (P1), both parents feed their offspring at a similar and roughly constant rate throughout the care period ('egalitarian care'). In (P2), one sex (here: the female) feeds at a maximal rate, while the other sex feeds at a much lower rate ('sex-biased care'). In (P3), one sex (here: the male) feeds again at a maximal rate, but the other sex does not feed the offspring at all ('uni-parental care'). In (P4), both sexes oscillate between provisioning at maximal and minimal rates; they do so in synchrony ('oscillatory care').

(P4) Oscillatory care. About 6% of our simulations resulted in a pattern where both sexes alternate between provisioning at maximal and minimal rates in strict synchrony. This pattern of biparental care results when the slopes of both reaction norms evolve to negative values ($\beta_m < 0$, $\beta_f < 0$) while the evolved values of $\alpha_m$ and $\alpha_f$ are similar ($\alpha_m \approx \alpha_f$).

## Efficacy of evolutionary outcomes

All six provisioning patterns described above evolved regularly, even though they differ in the total care provided to the offspring. In the simulations shown in Fig. 3B, the total provisioning rate is 1.22 (=0.58 + 0.64) in the case of egalitarian care (P1), 1.35 (=0.35 + 1.00) in the case of sex-biased care (P2), and 1.00 in the case of uniparental care (P3) and oscillatory care (P4). In line with this, offspring survival rates differ considerably between scenarios. In the sex-biased patterns, the sex that provisions at the maximal rate has considerably higher mortality than the sex that provisions at a lower rate (P2) or not at all (P3). Accordingly, the adult sex ratio becomes biased in favour of the less-caring sex, and many members of that sex cannot breed at all in a given season. Figure 4 illustrates that, as a result, the number of breeding pairs in the population is much smaller in the case of the biased care patterns (P2) and (P3) than in the unbiased care patterns (P1) and (P4). Notice that, in comparison to the baseline model (where the total provisioning rate fluctuated around 1.06), parental provisioning is higher in

negotiation equilibria of egalitarian care (P1) and sex-biased care (P2) and lower in uniparental care (P3) and oscillatory care (P4).

In all simulations, the populations were monomorphic for the evolved reaction norms. In other words, all members of a given sex employed the same sex-specific reaction norm (with the possible exception of a few rare mutants). Accordingly, all breeding pairs in the population showed a very similar provisioning pattern (Fig. 4).

## Evolutionary transitions between parental care patterns

It is well known that in stochastic systems with alternative stable states, rapid transitions between attractors will almost inevitably occur in a long-term perspective[16,17]. To assess the long-term stability of the provisioning patterns in our model, we ran 100 simulations for 1 million seasons. In 80% of these long-term simulations, the provisioning pattern did not change between seasons 40k and 1000k. However, evolutionary transitions from one care pattern to a different one occurred in 20% of these simulations. Figure 5 shows an example of a rapid transition from biparental care to female-biased care. This example is representative in the sense that, in our model, egalitarian biparental care is more 'labile' than the other forms of care: in all cases where an evolutionary transition occurred, it involved a switch from egalitarian care to either sex-biased care or uniparental care, while the opposite switch was never observed in our long-term simulations.

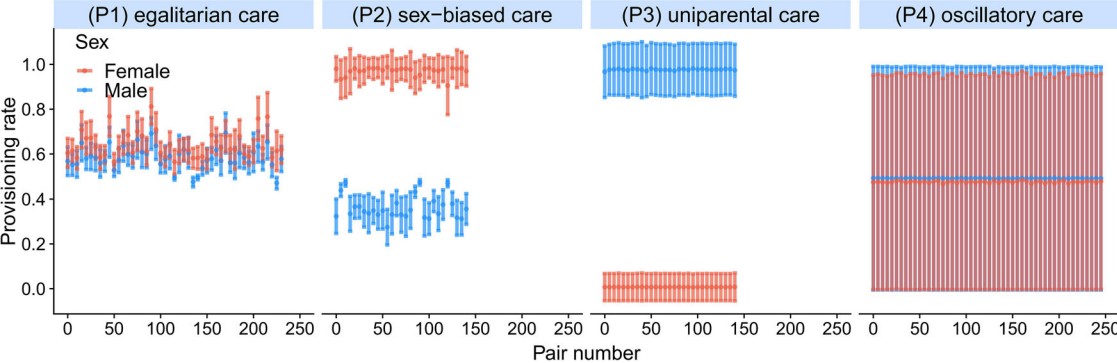

**Fig. 4 | Intrapopulation variation in parental provisioning patterns.** For each of the four simulations in Fig. 3B, representing the provisioning patterns (P1)–(P4), the provisioning rates (mean ± SD) of all breeding pairs in season 40k are shown (males in blue, females in red). (P1) Egalitarian care, where the males and females of all 230 breeding pairs provision their offspring at similar rates (0.58 and 0.64 feeds/min for the male and female, respectively) with limited variation within and across pairs throughout the season. (P2) Sex-biased care, where all females of the 140 breeding pairs provision at a much higher rate than their male partners. The higher breeding effort of the females results in a higher mortality and, consequently, in a male-biased population sex ratio. For this reason, the number of breeding pairs is much smaller than the maximum of 250. (P3) Uniparental care, where all males provision their offspring at the maximal rate, while all females feed at the minimal rate. Now, the population sex ratio is strongly female-biased. Accordingly, only 140 breeding pairs could be formed. (P4) Oscillatory care, where in all 245 breeding pairs the male and the female both oscillate between the maximal and the minimal provisioning rate. The simulations shown are representative in the sense that all simulations inspected exhibited very little within-population variation in offspring provisioning among breeding pairs.

### Robustness of our findings

To examine the robustness of the simulation outcomes, we conducted a comprehensive sensitivity analysis, as documented in Part 2 of the Supplement. We conducted numerous simulations for a larger population size, modified model parameters, and different shapes of the survival functions. All additional simulations confirm our main take-home message that the evolution of parental negotiation strategies can lead to a diversity of alternative parental care patterns. In all cases where the parameters or survival curves were the same for both sexes, we recovered the six attractors described above. However, as one would expect, the frequency distribution of evolutionary outcomes changed with a change in assumptions on, say, offspring needs or the effect of parental effort on parental survival (see Supplementary Table 1 and Supplementary Figs. 3-7 for details). Overall, egalitarian care is much less likely to emerge when the parental survival functions are convex-shaped and/or when offspring require a lower level of parental provisioning to survive.

### Discussion

To our knowledge, this study is the first attempt to investigate the evolution of parental negotiation strategies via individual-based simulations. In line with analytical models[13,28,41,42], we show that egalitarian biparental care can evolve, where both parents provision their offspring at a similar rate. However, egalitarian care with a constant provisioning rate is not the only evolutionary outcome. For the same parameter setting and initial conditions, almost 50% of the simulations converged to a qualitatively different evolutionary attractor. Even though the sexes have identical characteristics in our model, about 30% of the simulations led to sex-biased biparental care (P2) where either the male or the female provisions the offspring at a much higher rate than the other parent. More than 10% of the simulations ended up in uniparental care (P3) where only one of the parents does all the caring. A small percentage of the simulations ended up in an oscillatory equilibrium (P4), where both parents switch between periods with maximal and minimal parental effort. This seemingly strange outcome may reflect synchronised food provisioning in the wild, where parents time nest visits to coincide with each other's return[46,47]. In contrast to other individual-based simulation models[16,17,44], the evolved populations were monomorphic, that is, we did not observe systematic individual differences in negotiation strategies. In line with other models with alternative evolutionary attractors[16,17,48], the evolutionary outcome was 'labile' in the sense that, in a long-term perspective, rapid transitions between attractors occurred occasionally.

Many empirical studies have arrived at the conclusion that closely related species evolved very different patterns of parental care (from egalitarian care to sex-biased care to uniparental care[2,49]), and phylogenetic analyses revealed that switches from one care pattern to another one are surprisingly common[5,50,51]. Traditionally, this variation has been attributed to ecological factors such as food abundance[52], predation pressure[53], ambient temperature[54,55], or environmental stochasticity[56], to demographic factors like the sex ratio[57], and to sex differences related to life-history features[5] or sexual selection[49]. In line with such explanations, evolutionary transitions between parental care patterns have been attributed to a change in these factors[5,54,57,58]. Our study reveals that a diversity of care patterns can, in principle, evolve even in the same homogeneous environment and in the absence of pronounced sex differences in life history and mating patterns. This conclusion aligns well with recent phylogenetic studies, which find only a weak correlation between parental care patterns and the aforementioned factors[59] and with earlier simulation studies on a different type of parental care models[16,23]. Our model also illustrates that rapid transitions between parental care patterns can occur spontaneously, without any change in environmental or organismal conditions. Such transitions have also been reported earlier[16,17,23] but, to our knowledge, never in negotiation models or other models with a multivariate strategy space. As argued by Long and Weissing[16], spontaneous transitions between evolutionary attractors are less strange than they might appear; in fact, one should expect them to occur regularly (on a long-term perspective) whenever a system has alternative stable states and is subject to considerable stochasticity.

Alternative attractors, including attractors that break the symmetry between the sexes, have also been described in mathematical models for the interaction between the two sexes[16,60]. However, the focus of most mathematical parental care models was on finding an evolutionarily stable equilibrium corresponding to egalitarian biparental care. Various authors stated that equilibria corresponding to unisexual care might also exist, but, to our knowledge, the sex-biased negotiation equilibrium P2 has never been found or considered in mathematical models. In this equilibrium, the more-caring sex may be viewed as being exploited by the less-caring sex. This can be concluded from the slopes of the evolved reaction norms, which, in the case of P2, differ in sign (see Fig. 3). The more-caring sex employs a decreasing reaction norm ($\beta < 0$), indicating a compensatory negotiation rule ("increase your effort when the partner worked less than expected and decrease your effort otherwise"). One can imagine that such a rule can be exploited. In contrast, the less-caring sex employs an increasing reaction norm ($\beta > 0$), indicating a negotiation rule that resembles a 'tit-for-tat' kind

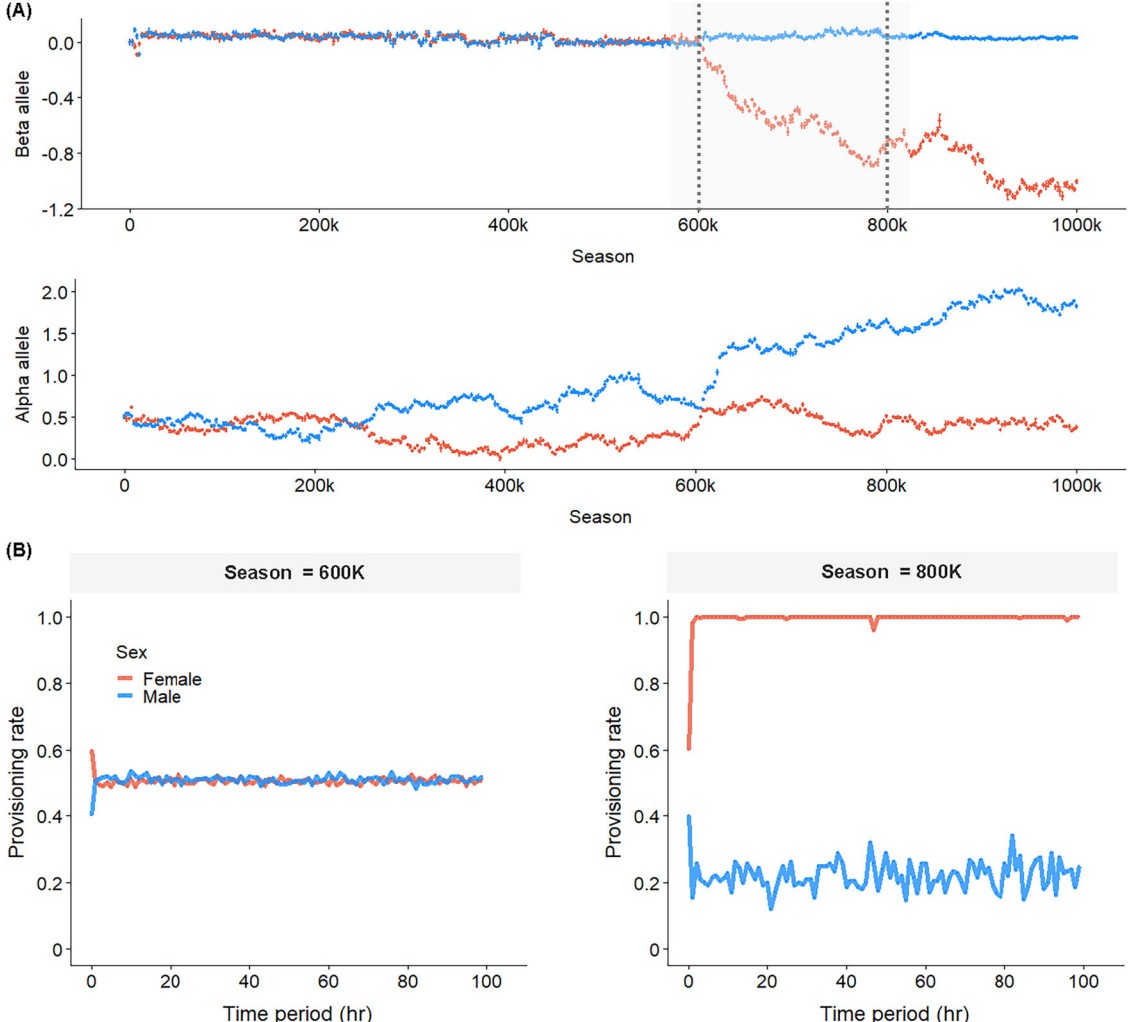

**Fig. 5 | Illustration of a rapid evolutionary switch between parental provisioning patterns. A** Long-term evolutionary dynamics of the allelic value ($\alpha_m, \beta_m, \alpha_f, \beta_f$) over one million (1000k) seasons. In the first half of the evolutionary trajectory, $\beta_m$ and $\beta_f$ are almost identical and slightly positive, leading to egalitarian parental care (P1). In this evolutionary phase, $\alpha_m$ and $\alpha_f$ differ from each other, but both stay below 1.0. Around season 600k, this pattern changes rapidly: $\beta_f$ becomes negative and $\alpha_m$ increases to much higher values, while $\beta_m$ and $\alpha_f$ more or less stay at their previous values. considerably, leading to a considerable difference between $\alpha_m$ and $\alpha_f$. After season 600k, $\beta_m$ is positive and $\beta_f$ is negative, leading to female-biased care (P2a). **B** Parental provisioning pattern in seasons 600k (egalitarian parental care) and 800k (female-biased parental care).

of strategy ("decrease your effort when your partner did not work hard enough and increase it otherwise"). Notably, the net effect is that the total level of parental care is higher in the case of sex-biased care (P2) than in the cases of egalitarian biparental care or uniparental care. However, in the sex-biased equilibrium P2, the exploitation of one sex ultimately backfires at the population level: the more caring sex becomes rarer due to increased mortality, which negatively impacts the mating prospects of the exploiting sex.

When negotiation models were first introduced, the general expectation was that negotiation tends to foster cooperation (as in ref. 45 and many economic models[61–63]). This expectation is supported by those of our simulations that lead to stable, egalitarian biparental care (P1), where both parents employ negotiation rules of the 'tit-for-tat' type. In these simulations, the total care level is lower than in P2 but higher than in the absence of negotiation (the baseline model). Interestingly, most mathematical studies[13,28,41,42] (but see ref. 36) arrive at the opposite conclusion that the total care level in an evolutionarily stable negotiation equilibrium is *lower* than in the corresponding model without negotiation (the Houston-Davies model[12]). In line with the mathematical model studies, our simulations suggest that in the case of uniparental care (P3), the total care level is lower than in the case of biparental care, irrespective of whether biparental care is

achieved with or without negotiation. However, we also regularly encountered the oscillatory biparental care equilibrium (P4), where the total care level does not exceed that of uniparental care.

It is well known from game theory that stochasticity can be of considerable importance for the evolutionary outcome[64]. Individual-based evolutionary models automatically incorporate several forms of stochasticity, such as demographic stochasticity, genetic drift, and the random influx of mutations, while most mathematical models only consider random mutations (but see refs. 38,65). Our model also considers another form of stochasticity, which arises from incomplete information on the previous behaviour of the interaction partner. In our model, the negotiation process is an integral part of the provisioning period (in line with ref. 41), and the idea is that in each step, the two parents respond to their partner's effort in the previous period. However, this effort may not easily be observable. If both parents are busy collecting food, the partner may be out of sight much of the time. Moreover, finding food is a stochastic process with a variable outcome for the same effort. Therefore, the partner's effort can only be deduced indirectly, for example, from the feeding state of the offspring. In our model, we implemented this by discriminating between the intended and the realised feeding rate of the partner and making strategic decisions dependent on the realised rates. In addition, our model assumes that negotiation

decisions are made on a longer timescale (time unit 'one hour') than the timescale of provisioning events (time unit 'one minute'). The choice of timescale is crucial because it determines the reliability of the information on which the negotiation decisions are based. If the timeframe is too short (say, after each provisioning event, as in ref. 36), the partner's feeding rate may reflect stochasticity (good or bad luck in finding food) more than the partner's effort, leading to a haphazard and jumpy negotiation. If the time scale is too long, it may take ages until a stable negotiation outcome is reached. In contrast to the more coarse-grained mathematical models, individual-based simulation models must include detailed assumptions on the exact timing of events and the information on which decisions are based. The necessity of including such detail is one reason why individual-based simulation approaches are not yet very popular in evolutionary biology.

We hope that our study shows that such approaches are nevertheless a valuable addition to the toolbox of evolutionary theory for at least three reasons. First, individual-based models include strategically important stochasticity (e.g., 'trembling hand stochasticity'[64]) in a natural and realistic manner. Second, the fact that the relationship between abstract strategies and concrete behaviours needs to be specified in detail prevents inconsistencies that are part of many mathematical (negotiation) models but are not easily spotted[41] (see the Supplementary Text for more information). Third, individual-based models are not based on the analysis of a fitness function but are explicitly dynamic, thus revealing the potential multiplicity of evolutionary outcomes and the existence of outcomes (like sex-biased care) that are easily overlooked otherwise.

Of course, simulation approaches also have their downside. To get a good overview of the range of dynamic behaviours within a reasonable time frame, the set of model parameters needs to be restricted. We consider the focus on a two-parameter family of reaction norms (characterised by the evolvable parameters $\alpha$ and $\beta$) the most important restriction of our model, as only increasing or decreasing functions of sigmoidal shape could evolve, thus limiting the flexibility of parental responses. It is possible that, due to this limitation, other negotiation outcomes (such as parental turn-taking[36]) never evolved. To mitigate the strategic restrictions, a broader family of reaction norms could be employed[44]. Alternatively, parental response strategies could be implemented via evolvable neural networks[48], which have limitless flexibility if the network topology is also allowed to evolve.

## Conclusions and empirical implications

In the parental care literature, it is often assumed that sex biases in parental care are the result of either sex differences in life history, morphology, and physiology or factors like sexual selection or uncertainty of parentage[66–68]. Apparently, researchers implicitly assume that asymmetries between the sexes are required to explain sex biases in parental behaviour. In line with this, theoreticians have, in models lacking such asymmetries, typically restricted their analysis to egalitarian evolutionary equilibria. Our study shows that such a restriction is not warranted. Evolutionary feedback between male and female parental behaviour can lead to the divergence of male and female strategies, ultimately resulting in asymmetric equilibria such as sex-biased or uniparental care[16,17].

Furthermore, classical theoretical and empirical work on parental negotiation often hinges on the idea that partial compensation (a negative slope $\beta$ of the reaction norm) is a necessary condition for the evolutionary stability of biparental care[9,13,28,34]. In contrast, our model shows that compensation destabilises parental cooperation: when a compensatory strategy evolves in only one of the sexes, sex-biased care results (P2) where the compensatory sex is exploited by the other sex; a compensatory strategy in both sexes either results in biparental care of low efficacy (P4) or uniparental care, which corresponds to the collapse of cooperation (P3). Conversely, biparental cooperation (P1) persists when a tit-for-tat negotiation rule (i.e. a reaction norm with a positive slope β) evolves in both sexes. This type of responsive rule aligns with empirical studies showing 'matching responses'[30,69] and supports recent theoretical and empirical research on reciprocal cooperation[35,36,38]. If both parents apply a tit-for-tat rule, this has a stabilising effect: even if one sex temporarily reduces its care contribution

(either voluntarily or accidentally), the system returns to the previous care level within a few interactions. In other words, biparental care is maintained at an efficacious equilibrium that is stabilised by mutual parental responsiveness. It is not clear to us why the mathematical analyses of parental care tend to conclude that parental compensation is a prerequisite for stable parental cooperation, while compensation destabilises cooperation in our model. Potential reasons for this discrepancy include (a) the use of a simple fitness function in the mathematical models, which neglects ecological and demographic processes that are included in simulation studies (see Supplementary Text 1); (b) the assumption of a cost-free negotiation phase in most mathematical models, while negotiation is costly in our model, as it is part of the provisioning period; (c) the 'trembling hand' stochasticity in our model, which can lead to very different evolutionarily stable outcomes than in models neglecting stochasticity[64,70], and (d) the fact that we considered a limited set of negotiation strategies, characterised by logistic functions, while a different and often larger strategy set is often considered in mathematical analyses.

The primary objective of our study was the conceptual clarification of the role of negotiation in the evolution of parental care. However, we hope that the approach taken and the findings made will open new avenues for empirical research. We welcome additional experimental studies designed to elucidate the nature of parental norms of reaction (as in refs. 31,71), ideally allowing comparisons between (phylogenetically diverse) species. As we have seen, focusing on sex differences in parental reaction norms may be particularly important[71,72]. The emphasis should not only be on parental compensation[13,34] but also on tit-for-tat kind of rules[36,38], allowing to test whether and to what extent the diversity of parental care patterns found in nature can be explained by a simple scheme as in Fig. 3A. Most likely, the negotiation rules in natural systems are more sophisticated than those in our conceptual model. Finding this out may be of considerable importance for understanding parental cooperation. Fortunately, individual-based models like the one presented here are highly flexible and can easily be expanded in response to new empirical findings.

## Methods

### Intended and perceived provisioning rates

As illustrated in Fig. 1A, the offspring provisioning period is subdivided into 100 time periods that we imagine to correspond to one hour each. Throughout the time period, the two parents have intended provisioning rates of $F_m$ and $F_f$ feeds per minute, respectively. The expected number of feeds per hour (60 min) is therefore $60 \cdot F_m$ for the male parent and $60 \cdot F_f$ for the female parent. As food search is a stochastic process, we assume that the actual provisioning rates in a given hour, $N_m$ and $N_f$, are random variables that are drawn from Poisson distributions with means $60 \cdot F_m$ and $60 \cdot F_f$. Hence, the realised provisioning rates per minute are $\widetilde{F}_m = \frac{N_m}{60}$ and $\widetilde{F}_f = \frac{N_f}{60}$. We assume that these realised rates can be perceived by the partner (e.g. on the basis of the feeding state of the offspring) and form the basis of the partner's decision regarding the intended provisioning rate in the following time period.

### Behavioural reaction norms

As illustrated in Fig. 1B, C, the parental food provisioning to their offspring is governed by inherited reaction norms (or negotiation rules). We consider reaction norms of the form:

$$F_{ego}^{t+1} = R(\widetilde{F}_{partner}^t) = \left(1 + \exp\left(-\beta \cdot \left(\widetilde{F}_{partner}^t - \alpha\right)\right)\right)^{-1} \quad (1)$$

In other words, ego's provisioning rate $F_{ego}^{t+1}$ in the following time period is determined by the partner's perceived provisioning rate $\widetilde{F}_{partner}^t$ in the previous time period via a logistic function $R$. The function $R$ is sigmoidal and ranges between zero and one; it is monotonically increasing if $\beta > 0$ and monotonically decreasing if $\beta < 0$ (see Fig. 1B, C). $R$ is characterised by the parameters $\alpha$ and $\beta$, which have the following interpretation: $\alpha$ corresponds to that provisioning rate of the partner that elicits a

provisioning rate of 0.5 ($F_m^{t+1} = 0.5$ if $\widetilde{F}_{partner}^{t} = \alpha$), and $\beta$ is proportional to the steepest ascent of the function $R$ ($R'(\alpha) = \frac{1}{4}\beta$). The individuals in our model are haploid and carry four genes with alleles ($\alpha_m, \beta_m, \alpha_f, \beta_f$), where $\alpha_m$ and $\beta_m$ are expressed if the individual is a male, whereas $\alpha_f$ and $\beta_f$ are expressed if the individual is a female. Hence, the reaction norm $R_m$ of the male parent is given by Eq. (1), with $\alpha$ and $\beta$ replaced by the male's inherited values $\alpha_m$ and $\beta_m$, while the reaction norm $R_f$ of the female is obtained from (1) by substituting the female's inherited values $\alpha_f$ and $\beta_f$.

## Parental provisioning dynamics

Given the reaction norms $R_m$ and $R_f$ and the intended provisioning rates in the first time period (which we assume to be $F_m^{t=1} = F_f^{t=1} = 0.5$ per minute for both parents), we can now deduce the provisioning sequence of both parents throughout the whole provisioning period. Based on the intended provisioning rates $F_m^t$ and $F_f^t$ for the time period $t$, we can first derive the realised provisioning rates $\widetilde{F}_m^t$ and $\widetilde{F}_f^t$. Using Eq. (1), this yields the intended provisioning rates for the next time period: $F_m^{t+1} = R_m(\widetilde{F}_f^t)$ and $F_f^{t+1} = R_f(\widetilde{F}_m^t)$. Averaging all realised provisioning rates yields a measure of the overall provisioning effort of the male and female parent:

$$E_m = \frac{1}{100} \cdot \sum_t \widetilde{F}_m^t \text{ and } E_f = \frac{1}{100} \cdot \sum_t \widetilde{F}_f^t \qquad (2)$$

## Effects of provisioning effort on parental survival

We assume that the survival probability to the next breeding season is for each parent negatively related to the parent's overall provisioning effort:

$$S_m = S(E_m) = 1 - \frac{1}{3} \cdot e^{E_m} \text{ and } S_f = S(E_f) = 1 - \frac{1}{3} \cdot e^{E_f} \qquad (3)$$

As the overall efforts $E_m$ and $E_f$ range between zero and one, the parental survival probabilities range between $S(0) = 2/3$ (in the case of no effort) and $S(1) \approx 0.1$ (in the case of maximal effort). In Part 2 of the Supplement, we also consider different parental survival functions.

## Effects of provisioning effort on offspring survival

We assume that the probability $S_{off}$ of an offspring to survive to the juvenile stage depends on the sum of the overall efforts of both parents, $E_{tot} = E_m + E_f$. In our model, it is given by the sigmoidal function

$$S_{off}(E_{tot}) = \frac{A \cdot E_{tot}^2}{E_{tot}^2 + B} \qquad (4)$$

Our parameter choice of $A = 1.30$ and $B = 1.15$ implies that the survival probability of an offspring is zero in the absence of parental provisioning ($S_{off}(0) = 0$), 60% if the total effort of both parents is $E_{tot} = 1.0$ ($S_{off}(1) = 0.6$), and 100% if $E_{tot} = 2.0$ ($S_{off}(2) = 1.0$). For a clutch size of $C$, the expected number of surviving offspring is $S_{off}(E_{tot}) \cdot C$. Here, we assumed $C = 5$ for all pairs, and the actual number of surviving offspring was drawn from a Poisson distribution with mean $S_{off}(E_{tot}) \cdot 5$. Although each pair reproduces only once in a breeding season, the total effort of both parents tended to be larger than or equal to one in all our simulations, three or more juveniles were produced on average in each breeding attempt. In the Supplement, different values of the parameter B are considered as well. In our model, the only costs of parental effort are in terms of parental survival. For simplicity, we neglect other costs, such as fewer opportunities for extra-pair matings or a reduced competitive ability on the mating market in the following season.

## Inheritance of behavioural reaction norms

When a new offspring is formed, its genotype ($\alpha_m, \beta_m, \alpha_f, \beta_f$) is formed by drawing each of the four alleles separately from the corresponding alleles of its parents, with a 50% probability from the father and a 50% probability from the mother. Hence, the four genes are not linked, and the same pair of parents can create offspring with $2^4 = 16$ possible genotypes. After assigning the offspring's genotype, mutations with a small effect size can occur. A mutation occurs at each of the four loci with probability $\mu = 0.01$. If a mutation occurs, the mutational step size is drawn from a normal distribution with a mean of zero and a standard deviation of $\sigma = 0.1$, and it is added to the allelic value inherited from one of the parents. Moreover, a new offspring is assigned a sex, where both sexes have the same probability.

## Population dynamics

We consider a population that, at the start of each breeding season, has a constant size of 500 individuals. Preliminary simulations indicated that this population size is sufficiently large for selection to dominate genetic drift. At the same time, it is small enough to run replicate evolutionary simulations over tens of thousands of breeding seasons within a reasonable time frame (Part 2 of the Supplement shows some results for a population size of 2000 individuals). A breeding season starts with pair formation, where males and females are matched randomly as long as unmated individuals of both sexes are still available. After pair formation, each pair produces a clutch of offspring and feeds their young, as explained above. Each individual has only one breeding opportunity per season. At the end of the season, all mated adults survive to the next breeding season with a probability given by Eq. (3). Unmated individuals have a survival probability of $S(0) = 2/3$. The offspring of the season survive to the juvenile stage with a probability given by Eq. (4). All surviving adults are transferred to the next season, while all non-surviving adults are removed from the population and replaced by (randomly drawn) offspring juveniles. The clutch size of five ensured that sufficiently many juveniles were produced for this replacement. Consequently, the population size at the start of the season remains constant, but the adult sex ratio may get biased, implying that not all individuals of the most abundant sex can breed in the next season.

## Simulation details

We initialised the population with 250 males and 250 females that all carried the allele combination ($\alpha_m, \beta_m, \alpha_f, \beta_f$) = (0.5, 0, 0.5, 0). Hence, initially all individuals employ a non-responsive strategy ($\beta = 0$) leading to a provisioning rate of 0.5 food items per minute throughout the whole provisioning phase. Each simulation was run for at least 40,000 ('40k') seasons, as this time was sufficient to produce evolutionarily stable provisioning patterns. However, we also ran very long simulations (for 10,000k seasons) to check for stability (e.g. Fig. 4). Of all shorter simulations (40k seasons), at least 100 replicate simulations were run to check for consistency in outcomes. At least 50 replicates were run for the ultralong (10,000k) simulations. The simulation code was written in C++ (version 22) and is freely available. Simulation data were analysed and visualised in R-(version 3.3.0+). This theoretical study did not require ethical approval.

## Baseline model

In the Supplement, we show simulation results of the baseline model where the (intended) parental provisioning rates $F_m$ and $F_f$ do not result from negotiation but are heritable parameters that remain constant throughout an individual's lifetime. The baseline model is identical in all aspects to the negotiation model described above, with one exception. Instead of harbouring four alleles ($\alpha_m, \beta_m, \alpha_f, \beta_f$) determining the male and female negotiation strategy, each individual now carries two alleles ($F_m, F_f$) determining the male and female provisioning rate. These alleles are also transmitted to the offspring in a Mendelian manner, subject to mutations as described above. However, $F_m$ and $F_f$ are restricted to the unit interval. Therefore, mutations with a negative value were replaced by zero, while mutations above one were replaced by one.

## Reporting summary

Further information on research design is available in the Nature Portfolio Reporting Summary linked to this article.

## Data availability

Simulation data that support the findings of this study have been deposited in FigShare[73]: https://doi.org/10.6084/m9.figshare.29665328.v1 Note that we do not analyse or generate any genuine datasets in the study, because our work proceeds within a theoretical and mathematical approach. Any remaining information can be obtained from the corresponding author upon reasonable request.

## Code availability

All model scripts used to generate the simulation data in this study have been deposited in FigShare[73]: https://doi.org/10.6084/m9.figshare.29665328.v1.

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

## Acknowledgements

We acknowledge funding from the National Natural Science Foundation of China (no. 32301290 to J.Z.); the European Research Council (ERC Advanced Grant no. 789240, to F.J.W.); the MSCA Seal of Excellence @UNIPD program (project 'CONFLICTnCARE' to D.B). We thank Zhengwang Zhang for commenting on the previous draft, and the two reviewers for their constructive peer review and valuable suggestions. We also thank the Centre for Information Technology of the University of Groningen for their support and for providing access to the Hábrók high-performance computing cluster.

## Author contributions

J.Z.: Designed, programmed and implemented the model, analysed and visualised simulation data, interpreted the results, drafted and revised the manuscript. F.J.W.: Developed the conceptual idea, supervised model development and data visualisation, interpreted the results, and led the manuscript revision. D.B.: Contributed to the conceptual framework and participated in manuscript writing and revisions.

## Competing interests

The authors declare no competing interests.
