## [Transparent Peer Review file · Communications Biology]

The evolution of negotiation strategies diversifies parental cooperation

Corresponding Author: Dr Jia Zheng

Version 0:

Reviewer comments:

Reviewer #1

(Remarks to the Author)

Mathematical studies of negotiation games between parents in relation to their respective investments made into their offspring have led to quite disparate results. The authors of the present study propose to make headway by performing an individual-based simulation analysis, to incorporate factors such as stochasticity and the possibility for polymorphism that have typically been assumed absent and to circumvent having to specify an explicit fitness function that may fail to capture relevant effects. They find that, even when parameter values and starting conditions are held fixed, different replicates of the evolutionary process may lead to qualitatively different outcomes, including equal parental investments, sex-biased investments (in both directions) and uniparental investment (again, in both directions); and when longer term evolution is considered they observe occasional transitions between each of these alternative regimes.

The simulations appear to have been carried out carefully, the assumptions and procedures are clearly stated, the simulation data are nicely analysed, and the paper is very well written. I find the results interesting and thought-provoking, and although I suspect that they are quantitatively dependent on the assumption of a population of only 500 individuals I do feel that qualitative points are likely to be robust.

Some scattered comments:

Line 43, awkward switch from present to past tense.

Lines 83-84, etc, why are forenames given for some authors but not others?

Line 520, "Smith JM" -> "Maynard Smith J"

Line 174, this made me eager to know exactly what the starting conditions were—although this information is provided later in the Methods, I think it would be helpful to say something about that somewhere around here.

Lines 256-265, and 343-344, this struck me as being directly analogous to the main message of Micheletti et al's (2018, Proc B 285, 20180975) analysis of sex differences in warfare (full disclosure: I'm one of the authors of that paper). I mention this just in case it is of interest to the present authors, and also to highlight that such hysteresis effects are not particular to individual-based simulations but may be reproduced in standard mathematical analyses (it might be helpful for the authors to emphasise this point).

As is my policy, I waive anonymity

Andy Gardner

Reviewer #2

(Remarks to the Author)

Overall, I am positive on "The evolution of negotiation strategies diversifies parental cooperation". The introduction is well written, clearly identifies an important gap in the literature, explains how the gap will be filled, and sets up an interesting

question. The results are interesting and well explained, although I am concerned that they are not thorough. The discussion is clear and raises many nice points, although there are opportunities for better connection with past literature. The model appears reasonable (but note that this was hard to assess in detail due to a typesetting error that rendered parameters non viewable).

1. My biggest concern is that the authors present their results with no sensitivity analysis for the role of various parameters. The claims that the authors make about the different evolved strategies are quite general and more work is needed to ensure that these conclusions are not sensitive to different parameter values. Examples of claims I am referring to come on lines 168-171 and 186. An example of the type of analysis that I feel addresses this concern comes on lines 413-420.

2. My second major comment is that the authors do not always clarify why differences between their model and past frameworks emerge. While noting these differences is nice, it is much more useful to also discuss what about the underlying biology causes such differences to emerge. See, in particular, lines 290-295 and 349-352.

Specific comments:

Lines 77-80: Exemplified by this sentence, the introduction seems to assume substantial prior knowledge about game-theoretic approaches to negotiation. The reader would benefit from a bit more explanation of how these negotiation models work. For instance, what is an example of a "negotiation rule" that can evolve in these models?

Paragraph starting on line 98: It is not completely clear what this shortcoming of (or alternative to) "the detailed analysis of the two parental fitness functions" is. I believe that the authors are referring to the lack of an explicit, dynamic model for evolution, but this could be made clearer.

Lines 118-119: A reminder of what the inconsistency is would be helpful.

Line 166 and throughout: Parameter symbols are missing from the text and are instead filled by question marks in boxes. This made a detailed review of the model quite challenging.

Lines 300-320: The assumptions regarding imperfect information and the timescale are critical but no information about this is provided in the model overview. Providing a brief note of these facts would help readers that do not carefully read the methods.

Line 320: Consider a paragraph break here.

Line 340: Should this be bolded as a new section header?

Lines 449-451: I am a bit confused about how offspring survival and recruitment works. I think more clarity is needed here and in the section of offspring survival. If I'm understanding, what the authors are referring to as offspring survival is really the opportunity for the offspring to survive to become an adult, whereas whether offspring actually survive to breed is determined by how many adults die (due to the assumption of fixed population size). These points could be made clearer, since the text from the subsection on lines 421 makes it sound like equation (4) completely determines offspring survival to adulthood. Also, what happens if fewer offspring are produced and survive than adults that die (so the population size cannot be returned to 500).

Methods: How many offspring are produced per pair each breeding season? Is it just the parameter C? This could be made clearer.

Figure S1 legend: The m and f from F_m and F_f should be subscripted.

Version 1:

Reviewer comments:

Reviewer #1

(Remarks to the Author)

The authors have fully addressed the concerns I raised in the previous round.

As is my policy, I waive anonymity

Andy Gardner

Reviewer #2

(Remarks to the Author)

The authors have done a very nice job addressing my comments. I believe that this will make a valuable contribution to the literature. I have only a few minor comments remaining for the authors to consider.

Table S1 is a really helpful addition. I understand that the authors don't want to spend much space on this, but I do think it would be valuable to highlight some of the parameters with the biggest effect (rather than simply saying that parameters matter on lines 229-231). For instance, it seems high offspring survival favors egalitarian care whereas convex care functions greatly disfavor egalitarian care.

One assumption that stood out to me on this reading is that the only cost to providing care is to survival. Of course, one could imagine that another cost is being unable to pursue extra pair copulations (likely a source of sex differences in care). It is of course outside the scope of this study to formally analyze such a dramatic change to the model. However, I wonder if the authors would like to briefly discuss the importance of this assumption and how relaxing it may change results.

Lines 198-208: It would be easier on the reader to use short descriptors like "egalitarian care" in place of the labels P1 – P4.

Detailed response to the reviewers' comments on COMMSBIO-25-4243:

“The evolution of negotiation strategies diversifies parental cooperation”

We thank the reviewers for their thoughtful and constructive comments. In response, we carefully revised our manuscript at various places. Most importantly, we added a comprehensive **sensitivity analysis** to the new version. To this end, we ran numerous additional simulations that are documented in (the new) Part 2 of the Supplement. More specifically,

- we increased the population size (in response to Reviewer 1);
- we considered different values for the most important parameters of the parental and offspring survival functions (in response to Reviewer 2);
- we considered parental survival functions with a different shape (in response to Reviewer 2);
- we considered a scenario where the male survival function differs in shape from the female survival function.

Supplementary Table S1 provides an overview of the outcome of the additional simulations, which are documented in more detail in **Supplementary Figures S2 to S7**. This material is too extensive to repeat here. In a nutshell, the eight additional scenarios considered all confirm our main conclusion that the evolution of negotiation strategies diversifies parental provisioning patterns. This even holds for the 8th scenario with a strong asymmetry between the sexes, although in this case, two of the six attractors were not observed in 100 replicate simulations. In the other scenarios, all six provisioning patterns (egalitarian care, female-biased care, male-biased care, female-only care, male-only care, oscillatory care) were evolutionary attractors as well. Not surprisingly, the relative frequency of the six patterns changed with a change in parameter values or a change in survival function shape. We have briefly summarised the above conclusions in a **new section** entitled “Robustness of our findings” in the main text (lines 223-231).

Reviewer 1:

Mathematical studies of negotiation games between parents in relation to their respective investments made into their offspring have led to quite disparate results. The authors of the present study propose to make headway by performing an individual-based simulation analysis, to incorporate factors such as stochasticity and the possibility for polymorphism that have typically been assumed absent and to circumvent having to specify an explicit fitness function that may fail to capture relevant effects. They find that, even when parameter values and starting conditions are held fixed, different replicates of the evolutionary process may lead to qualitatively different outcomes, including equal parental investments, sex-biased investments (in both directions) and uniparental investment (again, in both directions); and when longer term evolution is considered they observe occasional transitions between each of these alternative regimes.

The simulations appear to have been carried out carefully, the assumptions and procedures are clearly stated, the simulation data are nicely analysed, and the paper is very well written. I find the results interesting and thought-provoking, and although I suspect that they are quantitatively dependent on the assumption of a population of only 500 individuals I do feel that qualitative points are likely to be robust.

#R1.1: Thanks for this positive evaluation. As to the population size, we typically ran 100 replicate simulations, each simulation running over 40k or more seasons, and within a season, all mated pairs had numerous interactions. To limit the simulation time, we chose a relatively small population size of 500 individuals. However, we had checked before that our results did not change substantially in comparison to those in much larger populations. This is now documented in Supplementary Table S1 and Figure S3, Lines 438-439. There, we show that the same six attractors are also observed in populations of 2,000 individuals, and that the provisioning patterns and the evolution of negotiation strategies in the larger populations are almost identical to those in the smaller populations. Interestingly, the percentage of simulations resulting in egalitarian care was larger for 2,000 individuals than for 500 individuals.

Some scattered comments:

Line 43, awkward switch from present to past tense.

#R1.2: Corrected.

Lines 83-84, etc, why are forenames given for some authors but not others?

#R1.3: Thanks for spotting this. Here and elsewhere, we removed the forenames.

Line 520, "Smith JM" -> "Maynard Smith J"

#R 1.4: Of course! Corrected.

Line 174, this made me eager to know exactly what the starting conditions were—although this information is provided later in the Methods, I think it would be helpful to say something about that somewhere around here.

#R 1.5: Line 174 referred to Fig. 2, and the legend of this figure specifies the starting conditions as follows: "All simulations were initialised at $\alpha_m = \alpha_f = 0.5$ and $\beta_m = \beta_f = 0.0$ and run for 40,000 seasons. The initial conditions correspond to a population where males and females provision their offspring at a constant rate of 0.5...". We believe this contains enough information of starting conditions for readers to comprehend.

Lines 256-265, and 343-344, this struck me as being directly analogous to the main message of Micheletti et al's (2018, Proc B 285, 20180975) analysis of sex differences in warfare (full disclosure: I'm one of the authors of that paper). I mention this just in case it is of interest to the present authors, and also to highlight that such hysteresis effects are not particular to individual-based simulations but may be reproduced in standard mathematical analyses (it might be helpful for the authors to emphasise this point).

#R1.6: Thanks for this comment. We cited the paper you mentioned. The third paragraph of the Discussion now starts with the sentence: “Alternative attractors, including attractors that break the symmetry between the sexes, have also been described in mathematical models for the interaction between the two sexes^{16, 60}. However...”(lines 267-268)

Reviewer #2:

Overall, I am positive on “The evolution of negotiation strategies diversifies parental cooperation”. The introduction is well written, clearly identifies an important gap in the literature, explains how the gap will be filled, and sets up an interesting question. The results are interesting and well explained, although I am concerned that they are not thorough. The discussion is clear and raises many nice points, although there are opportunities for better connection with past literature. The model appears reasonable (but note that this was hard to assess in detail due to a typesetting error that rendered parameters non viewable).

1. My biggest concern is that the authors present their results with no sensitivity analysis for the role of various parameters. The claims that the authors make about the different evolved strategies are quite general and more work is needed to ensure that these conclusions are not sensitive to different parameter values. Examples of claims I am referring to come on lines 168-171 and 186. An example of the type of analysis that I feel addresses this concern comes on lines 413-420.

#R2.1: We fully agree and thank the reviewer for this comment. As indicated above, we have now conducted a comprehensive sensitivity analysis that is documented in Part 2 of the Supplement (see Table S1 and Figures S2 to S7). A summary of this analysis has been added to the Results section of the main text (lines 223-231).

“Robustness of our findings. To examine the robustness of the simulation outcomes, we conducted a comprehensive sensitivity analysis, as documented in Part 2 of the Supplement. We ran numerous simulations for a larger population size, modified model parameters, and different shapes of the survival functions. All additional simulations confirm our main take-home message that the evolution of parental negotiation strategies can lead to a diversity of alternative parental care patterns. In all cases where the parameters or survival curves were the same for both sexes, we recovered the six attractors described above. However, as one would expect, the frequency distribution over the evolutionary outcomes changed with a change in assumptions on, say, offspring needs or the effect of parental effort on parental survival (see Table S1 and Figures S3 to S7 for details).”

2. My second major comment is that the authors do not always clarify why differences between their model and past frameworks emerge. While noting these differences is nice, it is much more useful to also discuss the underlying biological causes of these differences. See, in particular, lines 290-295 and 349-352.

#R2.2: Again, we are thankful for this comment, although it is not straightforward to provide a definitive answer. We have now added the following sentences to the final section of the Discussion (lines 355-364):

“It is not clear to us why the mathematical analyses of parental care tend to conclude that parental compensation is a prerequisite of stable parental cooperation, whereas compensation destabilises cooperation in our model. Potential reasons for this discrepancy include (a) the use of a simple fitness function in the mathematical models, which neglects ecological and demographic processes that are included in simulation studies (see Suppl. Text 1); (b) the assumption of a cost-free negotiation phase in most mathematical models, while negotiation is costly in our model, as it is part of the provisioning period; (c) the ‘trembling hand’ stochasticity in our model, which can lead to very different evolutionarily stable outcomes than in models neglecting stochasticity ^{64, 70}; and (d) the fact that we considered a limited set of negotiation strategies, characterised by logistic functions, while a different and often larger strategy set is often considered in mathematical analyses.”

Specific comments:

Lines 77-80: Exemplified by this sentence, the introduction seems to assume substantial prior knowledge about game-theoretic approaches to negotiation. The reader would benefit from a bit more explanation of how these negotiation models work. For instance, what is an example of a “negotiation rule” that can evolve in these models?

#R2.3: We have now rewritten this part as follows in lines 56-63:

“McNamara and colleagues ^{9, 28} argued that, in organisms with extended periods of biparental care, parental games should be modelled as involving a series of interactions in which the parents negotiate their mutual parental effort via inherited rules that prescribe how to respond to the behaviour of the other parent. A negotiation rule might, for example, prescribe the reduction of one’s own effort whenever the partner shows less effort than anticipated (as in the Tit-for-Tat strategy). Alternatively, it might prescribe compensatory behaviour that enhances an individual’s own effort in cases where the partner exhibits a reduced effort. According to this view, models should focus on the evolution of such rules instead of the evolution of fixed levels of care ²⁸. This approach is indeed more realistic, as there is ample evidence that caregivers respond to each other’s behaviour in nature ^{29, 30, 31, 32, 33}.”

Paragraph starting on line 98: It is not completely clear what this shortcoming of (or alternative to) “the detailed analysis of the two parental fitness functions” is. I believe that the authors are referring to the lack of an explicit, dynamic model for evolution, but this could be made clearer.

R2.4: Thank you for this comment. A full clarification of the “shortcomings” of the fitness function approach cannot be achieved in a few lines. We therefore devoted Part 1 of the Supplement to this matter. First, we show that the standard assumption

that “fitness” corresponds to the difference between reproductive benefits and reproductive costs is problematic, as it does not account for density regulation (which is fitness-relevant). Second, we derive a fitness function for our individual-based simulation model, where density is regulated via the limited number of adult breeding positions. It turns out that this function is not given by the difference between benefits and costs, but by a quotient. Third, we demonstrate (in Figures S1 and S2) that the predictions based on our fitness function agree reasonably well with the simulations (at least in the baseline version of the model). However, we also show that there is still a systematic bias in the predictions, which most likely is caused by the fact that our fitness function (as well as the standard fitness function used in the literature) does not take sex differences (due to deviations from a 1:1 sex ratio) into consideration. Most of the discussed “intricacies” are not generally known; therefore, we think that Part 1 of the Supplement is a valuable contribution to the literature. As it is too extensive to be included in the main text, we have added a pointer to it on lines 84-88:

“To keep the mathematical analysis tractable, many simplifying assumptions must be made that can strongly affect the model outcome. For example, analyses typically take a specific type of fitness function as their point of departure, without deriving this function from biological considerations. As explained in Supplementary Text 1, the most popular fitness functions (because of their mathematical convenience) are often not consistent with ecological or demographic considerations. Moreover,…”

Lines 118-119: A reminder of what the inconsistency is would be helpful.

#R2.5: We have now replaced the sentence with “Lessells and McNamara⁴¹ noted an inconsistency in the early negotiation models, as they did not specify the relationship between negotiation and provisioning.” (lines 90-91)

Line 166 and throughout: Parameter symbols are missing from the text and are instead filled by question marks in boxes. This made a detailed review of the model quite challenging.

#R2.6: Sorry for the inconvenience! When we translated our Word file to PDF, all equations looked OK. We hope the equations are visible for you in this revised version. Otherwise, we can provide you with our PDF version via the Editorial Office.

Lines 300-320: The assumptions regarding imperfect information and the timescale are critical but no information about this is provided in the model overview. Providing a brief note of these facts would help readers that do not carefully read the methods.

#R2.7: We agree and revised a sentence in the model overview as (lines 135-136) “The offspring provisioning phase is subdivided into discrete 1-hour time periods, within which provisioning behaviour occurs on a minute-level timescale. At the start of each time period,…” This aspect is treated in more detail in the Methods section.

Line 320: Consider a paragraph break here.

#R2.8: Done.

Line 340: Should this be bolded as a new section header?

#R2.9: Done. Line 333

Lines 449-451: I am a bit confused about how offspring survival and recruitment works. I think more clarity is needed here and in the section of offspring survival. If I'm understanding, what the authors are referring to as offspring survival is really the opportunity for the offspring to survive to become an adult, whereas whether offspring actually survive to breed is determined by how many adults die (due to the assumption of fixed population size). These points could be made clearer, since the text from the subsection on lines 421 makes it sound like equation (4) completely determines offspring survival to adulthood. Also, what happens if fewer offspring are produced and survive than adults that die (so the population size cannot be returned to 500).

#R2.10: Thanks for spotting this gap in our explanation. We now clarify matters by stating that the offspring survival function specifies survival to the juvenile stage (rather than survival to adulthood) "We assume that the probability S_{off} of an offspring to survive to the juvenile stage depends on the sum of the overall efforts of both parents..." (lines 414-415)

Later, we state that open positions in the adult population are filled by randomly chosen juveniles. The clutch size of five ensured that sufficiently many juveniles were produced for this replacement "...while all non-surviving adults are removed from the population and replaced by (randomly drawn) offspring juveniles." (lines 446-448)

We also argue that, given our clutch size of 5, on average more than 3 juveniles are produced per breeding attempt, This explains why always sufficiently many juveniles are available for filling open positions in the adult population. "Although each pair reproduce only once in a breeding season, the total effort of both parents tended to be larger than or equal to one in all our simulations, three or more juveniles were produced on average in each breeding attempt." (lines 422-425).

Methods: How many offspring are produced per pair each breeding season? Is it just the parameter C? This could be made clearer.

#R2.11: The clutch size is indeed $C = 5$ for all breeding pairs. We have the impression that the added sentence on the average number of juveniles produced per breeding attempt will clarify this matter to the reader in lines 422-423

"Although each pair reproduce only once in a breeding season, the total effort of..."

Figure S1 legend: The m and f from F_m and F_f should be subscripted.

#R2.12: Done.

Response letter COMMSBIO-25-4243B

Dear editor Dr. Rupali Sathe,

We are glad that our work, entitled "The evolution of negotiation strategies diversifies parental cooperation" has been accepted by Communications Biology! We appreciate the professional manuscript review by the two reviewers and the editors. Here in this letter, you can find our replies to the final comments.

Kind regards,

Jia Zheng, Franz J Weissing and Davide Baldan

REVIEWERS' COMMENTS:

Reviewer #1 (Remarks to the Author):

The authors have fully addressed the concerns I raised in the previous round.

As is my policy, I waive anonymity

Andy Gardner

Response: Glad to hear. Thank you for the professional review of our work!

Reviewer #2 (Remarks to the Author):

The authors have done a very nice job addressing my comments. I believe that this will make a valuable contribution to the literature. I have only a few minor comments remaining for the authors to consider.

Response 1: Thanks for your review and the positive comments! You helped us improve this work.

Table S1 is a really helpful addition. I understand that the authors don't want to spend much space on this, but I do think it would be valuable to highlight some of the parameters with the biggest effect (rather than simply saying that parameters matter on lines 229-231). For instance, it seems high offspring survival favors egalitarian care whereas convex care functions greatly disfavor egalitarian care.

Response 2: We have presented detailed result comparisons between the main model and specific model variants in Figure S3-S7. However, we also understand that a more straightforward general conclusion in the main text can be helpful. Therefore, we added a sentence in line 233-235: "Overall, egalitarian care is much less likely to emerge when the parental survival functions are convex-shaped and/or when offspring require a lower level of parental provisioning to survive."

One assumption that stood out to me on this reading is that the only cost to providing care is to survival. Of course, one could imagine that another cost is being unable to pursue extra pair copulations (likely a source of sex differences in care). It is of course outside the scope of this study to formally analyze such a dramatic change to the model. However, I wonder if the authors would

like to briefly discuss the importance of this assumption and how relaxing it may change results.

Response 3: Thanks for your comments. We agree that extra-paired copulation is likely to affect the model results. However, in our model, we purposely focus on the situation without external sex differences, and highlight that behavioural diversification can readily emerge under a sex identical condition. Therefore, we prefer to add an extra sentence at the end of the method session "Effects of provisioning effort on offspring survival", please see lines 430-432:

“For simplicity, we neglect other costs, such as fewer opportunities for extra-pair matings or a reduced competitive ability on the mating market in the following season.”

Lines 198-208: It would be easier on the reader to use short descriptors like "egalitarian care" in place of the labels P1 – P4.

Response 4: We have added descriptors at the proper places (Line 200-209)